# Motor Cortical Plasticity Induced by Volitional Muscle Activity-Triggered Transcranial Magnetic Stimulation and Median Nerve Stimulation

**DOI:** 10.3390/brainsci12010061

**Published:** 2021-12-31

**Authors:** Pramudika Nirmani Kariyawasam, Shinya Suzuki, Susumu Yoshida

**Affiliations:** 1Department of Nursing, Faculty of Allied Health Sciences, University of Ruhuna, Karapitiya, Galle 80000, Sri Lanka; pramudika@ahs.ruh.ac.lk; 2Department of Physical Therapy, School of Rehabilitation Sciences, Health Sciences University of Hokkaido, Tobetsu-cho, Ishikari-gun, Hokkaido 061-0293, Japan; suzukis@hoku-iryo-u.ac.jp

**Keywords:** bilateral training, transcranial magnetic stimulation, functional electrical stimulation, neuroplasticity

## Abstract

Bilateral motor training is a useful method for modifying corticospinal excitability. The effects of bilateral movement that are caused by artificial stimulation on corticospinal excitability have not been reported. We compared motor-evoked potentials (MEPs) of the primary motor cortex (M1) after conventional bilateral motor training and artificial bilateral movements generated by electromyogram activity of abductor pollicis brevis (APB) muscle-triggered peripheral nerve stimulation (c-MNS) and transcranial magnetic stimulation of the ipsilateral M1 (i-TMS). A total of three protocols with different interventions—bilateral finger training, APB-triggered c-MNS, and APB-triggered i-TMS—were administered to 12 healthy participants. Each protocol consisted of 360 trials of 30 min for each trial. MEPs that were induced by single-pulse TMS, short-interval intracortical inhibition (SICI), and intracortical facilitation (ICF) that were induced by paired-pulse TMS were assessed as outcome measures at baseline and at 0, 20, 40, and 60 min after intervention. MEP amplitude significantly increased up to 40 min post-intervention in all protocols compared to that at the baseline, although there were some differences in the changing pattern of ICF and SICI in each protocol. These findings suggest that artificial bilateral movement has the potential to increase the ipsilateral cortical excitability of the moving finger.

## 1. Introduction

The most common motor impairment that is seen among stroke survivors is paresis. Approximately 80% patients have upper arm paresis at the acute stage and 40% continue to have chronic upper arm paresis [1]. Upper arm paresis limits the activities of daily living (ADL) [2]. Older adults frequently use bilateral hand movements to perform everyday activities [3]; however, patients with stroke minimally use their paretic hand in daily life despite significant improvements in their hand function [4]. Non-use of the paretic hand causes a functional deficit in the nervous system. A contributing factor is an abnormal increase in inter-hemispheric inhibition (IHI) in the unaffected hemisphere [5]. Bilateral motor training (BMT) is an effective method for improving post-stroke upper extremity function and the ability to perform ADL [6]. Reviews on BMT have also emphasized its importance in improving upper arm function in patients with chronic stroke [7,8]. BMT induces a short-term increase in corticospinal excitability after training [8,9]. There is a balanced IHI between the two hemispheres in healthy individuals. However, in stroke, there is a disproportionate amount of inhibition from the contralesional hemisphere toward the ipsilesional hemisphere [10]. A hypothetical mechanism of recovery with BMT is the normalization of abnormal IHI [11]. Moreover, bilateral training using non-invasive brain stimulation (NIBS) has been investigated as a method to modulate cortical excitability and correct abnormal IHI [12]. A recent review outlined that NIBS could be used to modulate prepotent ongoing motor actions in several brain areas of healthy individuals, including the pre-supplementary motor area and the inferior frontal gyrus, which are associated with underlying inhibitory mechanisms [13].

However, because of the necessity of voluntary movement of the paretic limb, BMT is not indicated for patients with severe paresis. Moreover, the effectiveness of BMT is limited when patients have severe upper arm paresis as the effectiveness of the intervention depends on the level of severity [8]. For such patients, several technologies producing artificial movement of the paretic limb may facilitate the performance of BMT. Hence, it is important to study methods that can be used to artificially move the paretic arm. The voluntary movement of the non-paretic hand and artificial movement of the paretic hand is an alternative method to produce BMT. Therefore, NIBS, which modulates the excitability of the motor cortex, can be used to produce artificial movements of the paretic arm [14]. Electrical stimulation (ES) and transcranial magnetic stimulation (TMS) are the two widely used NIBS techniques for stroke rehabilitation [15]. ES of the peripheral nerves to innervate the muscles is a method that is used to easily produce artificial movements. Peripheral nerve ES activates the motor fibers and elicits muscle activities. Further, the sensory fibers in the same mixed nerve bundle cause afferent volleys and propagate to the sensory cortex. Additionally, the sensory fibers arising from the invoked muscle contraction also propagate to the sensory cortex. These ascending volleys seem to affect the motor cortex via the cortico–cortical network [16]. Methods for triggering contralateral limb movements have been used to assist BMT. Knutson et al. developed a protocol for contralaterally controlled functional ES (FES), and they investigated the effectiveness of improving hand function in patients with stroke [17]. Furthermore, contralaterally controlled FES in patients with hemiparesis showed a significant reduction in hand impairment after training sessions [18]. However, previous studies have used ES with voluntary contraction of the paretic arm. Therefore, it is unclear whether artificial bilateral movements have the potential to change the cortical excitability of the resting hemisphere.

Another way to create artificial movements is TMS over the motor cortex [19]. TMS activates neurons that are oriented horizontally in a plane that is parallel to both the coil and brain surface. This stimulation induces descending volleys in the corticospinal tract neurons that project on spinal motoneurons and it evokes muscle activity [20]. Some studies have used voluntary contraction or movement-triggered stimulation methods for TMS timing [21,22]. However, the effects of artificial bilateral movement on cortical excitability through TMS have not been reported. These artificial movement protocols will be helpful for motor recovery in patients with severe arm paresis as they are unable to move both arms voluntarily. Our hypothesis is that the artificial bilateral movements that are generated by FES and TMS in healthy subjects will induce motor cortex excitability that is comparable to that which is induced by voluntary bilateral training. To test our hypothesis, a voluntary bilateral movement protocol was used as a control test with artificially generated bilateral training protocols. Subsequently, voluntary bilateral training protocols were compared with voluntary muscle contraction-triggered ipsilateral TMS and voluntary muscle contraction-triggered contralateral median nerve stimulation (MNS) protocols. We expected a long-lasting increase in MEPs as our main outcome measure in all three protocols. Furthermore, short-interval intracortical inhibition (SICI) and intracortical facilitation (ICF) were measured to study the physiological features of cortical organization. The paired pulse TMS paradigm was used to measure SICI and ICF, where a test stimulus was preceded by a conditioning stimulus. Interstimulus intervals of approximately 1–5 and 6–10 ms caused SICI and ICF, respectively [23].

## 2. Materials and Methods

### 2.1. Participants

A total of 12 right-handed healthy adult participants (eight men and four women; age range, 20–50 years; mean age, 26 ± 8 years) without any neurological diseases were included after obtaining their written informed consent. In this study, the general stimulus procedures were performed in accordance with an updated report on the safety of TMS and peripheral nerve stimulation by the International Federation of Clinical Neurophysiology Committee [24]. The participants were seated comfortably on a chair and their forearms and wrists on both sides were fixed on a table in a neutral position during the experiments.

### 2.2. Study Design

The general experimental conditions and time course are shown in Figure 1. This study had a counterbalanced crossover design which consisted of three experimental sessions with different interventions—(1) bilateral finger movement training (BFT) involving bilateral thumb abduction; (2) electromyographic (EMG) activity of the right abductor pollicis brevis (APB)-triggered TMS of the ipsilateral M1 (APB-triggered i-TMS); and (3) EMG activity of the right APB-triggered contralateral median nerve stimulation (APB-triggered c-MNS) (Figure 1A). The intervention consisted of two blocks that lasted for 15 min each (see Section 2.6. for details). A break period of 5 min was interposed between the blocks. The outcome measurements were performed before (baseline), immediately after (at 0 min), and 20, 40, and 60 min after the intervention (Figure 1B). Each session lasted for approximately 2 h and it was performed on separate days with a gap of at least 1 week between the sessions.

### 2.3. Electromyography

Surface EMGs were recorded bilaterally from the APB. A pair of Ag/AgCl disc electrodes (NE-101; Nihon Kohden, Tokyo, Japan) was placed, with the active electrode placed over the muscle belly and the reference electrode placed over the metacarpophalangeal joint of the thumb. The EMG signals were amplified (×1000) and the band-pass was filtered (5–3000 Hz) through a bioamplifier (BIOTOP 6R12; NEC San-ei Instruments, Tokyo, Japan). The analog EMG signals were digitized at 6 kHz and stored on a computer using an A/D converter (Power 1401-3; Cambridge Electronic Design, Cambridge, UK) and data acquisition software (Spike2 version 7; Cambridge Electronic Design, Cambridge, UK).

### 2.4. TMS

A monophasic single-pulse TMS was administered using a magnetic stimulator (Magstim 2002; Magstim, Whitland, UK) and a figure-eight coil (D70 Alpha B.I.; Magstim). For the paired-pulse TMS protocol, a set of two magnetic stimulator units through a BiStim connecting module (Magstim BiStim2; Magstim) was used. The coil was held over the right scalp 45° lateral to the midsagittal line to ensure that the induced current flowed through the brain from the posterior to the anterior [25,26]. The optimal coil position to produce an MEP in the left APB was carefully determined in each session using the following procedure. Prior to data collection, a suprathreshold TMS was applied at various sites around the C4 of the international 10–20 system with the coil position moved in the anterior–posterior and medio–lateral directions by 1-cm step while the participants were at rest. The position where the TMS elicited 2–3 reproducible large MEPs in the left APB was defined as the left APB hotspot and it was marked on a swimming cap covering the scalp [27]. The resting motor threshold (RMT) was defined as the minimum intensity that produced an MEP of >50 µV in the left APB in at least 5 of 10 consecutive TMS pulses at 0.2 Hz while the participants were at rest. The RMT was determined by increasing or decreasing the stimulus intensity in steps of 1% of the maximum stimulator output.

### 2.5. Median Nerve Stimulation

A single rectangular electrical pulse (1 ms) was administered with an electrical stimulator (SEN-8203; Nihon Kohden) that was connected to a constant-voltage isolator unit (SS-104J; Nihon Kohden). The left median nerve at the wrist was stimulated using a pair of surface electrodes (NE-101; Nihon Kohden) with a bipolar montage (2 cm apart, cathode on the proximal). The optimal electrode positions for eliciting a large motor (M-) wave in the left APB were determined and the electrodes were fixed with elastic surgical tape.

### 2.6. Intervention

As mentioned above, each intervention included two blocks of 180 trials (360 trials in total). In the BFT protocol, the participants were requested to perform ballistic voluntary abduction movements of both thumbs simultaneously with maximum effort in response to an auditory imperative cue (tone burst, 1 kHz, 100 ms). A warning cue (tone burst, 500 Hz, 100 ms) was presented 0.8–1.5 s prior to the imperative cue to maintain arousal. The set of warning and imperative cues were presented at intervals of 5 s. We performed a training session before the ideal conditioning session, checked the MVC of quick thumb abduction, and set the target line at the MVC level. Visual feedback of the rectified and smoothed EMG signals was provided on a monitor in front of the participants and they were asked to contract the muscle at a constant force to maintain the EMG wave at the target line. In the APB-triggered i-TMS and APB-triggered c-MNS protocols, the participants were asked to perform a ballistic voluntary abduction movement of the right thumb alone and keep the left APB relaxed during the intervention. In these protocols, a specific EMG waveform of the right APB was discriminated in real time using a template-matching algorithm of a spike detector (Alpha Spike Detector; Alpha Omega Engineering, Nazareth, Israel). Template matching techniques were used to detect the muscle activity to prevent malfunction due to artifacts. Before intervention, the subjects were required to perform abduction of the right thumb a few times and the EMG was recorded. The EMG was sorted by the spike shape using the real-time sorting algorithm [28] and the typical shapes that appeared frequently were converted into a transistor–transistor logic (TTL) pulse event. Only a single-pulse event was generated per movement to prevent unnecessary high-frequency stimulation which would cause muscle fatigue or other undesirable effects. Then, the generated TTL pulse triggered TMS over the right M1 in the APB-triggered i-TMS and electrical stimulation of the left median nerve in the APB-triggered c-MNS. The time lag between the detection of the EMG by the algorithm and that sent out of the TTL pulse was 2 ms. The time between the spikes for evoking muscle activity was 21.57 ± 1.24 ms and 3.56 ± 0.18 ms for TMS and MNS, respectively. The stimulus intensity of TMS was set at 120% of the RMT. The stimulus intensity of the MNS was set at 120% of the stimulator output that was required to elicit the maximum M-wave.

### 2.7. Outcome Measures

There were four different outcomes that were measured—RMT, MEP amplitudes, SICI, and ICF. All the measurements were performed at rest. To measure the MEP amplitude, 10 MEPs were evoked by a train of single TMS pulses at 0.17–0.25 Hz [29,30,31]. The TMS intensity that elicited an MEP of 0.5–1 mV was determined at baseline and the intensity was kept constant across time periods. A test-conditioning stimulation paradigm was used to measure SICI and ICF. A total of 10 test MEPs and 10 conditioned MEPs were elicited by randomly altered single (for test MEPs) and paired (for conditioned MEPs) TMS pulses at 0.17–0.25 Hz [23]. The stimulus intensity of the test TMS pulse was set at an intensity that evoked an MEP of 0.5–1 mV and adjusted at each time period when necessary. The stimulus intensity of the conditioning TMS pulse was set at 0.8 × RMT [23]. Interstimulus intervals between the test and conditioning pulses were set at 2 ms for SICI and 10 ms for ICF.

### 2.8. Data Analysis

In the offline analysis, the peak-peak MEP amplitude was measured in individual unrectified EMG sweeps. The mean peak-peak amplitude across 10 sweeps was then calculated. The MEP amplitude for single-pulse TMS was normalized to the baseline value. For SICI and ICF, the amplitude of the conditioned MEP was normalized to that of the MEP test.

All statistical tests were performed using SPSS software (SPSS Statistics version 25; IBM, Chicago, IL, USA). Two-way repeated measures analysis of variance (ANOVA) was performed for each dependent variable (for the MEP amplitude, intervention 3 levels × time 4 levels; for RMT, SICI, and ICF, intervention 3 levels × time 5 levels). If the reported F value was statistically significant, a post hoc test was performed with Tukey’s test to reveal differences from the baseline values. A separate one-way ANOVA was conducted of the baseline MEP and RMT to identify whether there was an effect of intervention (intervention 3 levels). In addition, the effect size (partial eta squared: η_p_^2^) and the observed power (1 − β) were also computed for each ANOVA factor. Statistical significance was set at *p* < 0.05. The group data are shown as the mean ± standard error of the mean.

### 2.9. Ethical Approval

This study was conducted in accordance with the principles of the Declaration of Helsinki. Ethical approval was obtained from the Ethics Review Committee of the School of Rehabilitation Sciences, Health Sciences University of Hokkaido (approval number: 18R057066).

## 3. Results

### 3.1. RMT and MEP Amplitude

Figure 2A shows the average RMT. The two-way ANOVA showed no significant effect of intervention (F (2, 22) = 2.79, *p* = 0.08, η_p_^2^ = 0.202, 1 − β = 0.492) and time (F (4, 44) = 0.94, *p* = 0.45, η_p_^2^ = 0.079, 1 − β = 0.274) or interaction [F (8, 88) = 0.84, *p* = 0.57, η_p_^2^ = 0.071, 1 − β = 0.366]. For the RMT at baseline, a separate one-way ANOVA revealed a significant effect of intervention (F (2, 22) = 3.448, *p* = 0.05, η_p_^2^ = 0.239, 1 − β = 0.585). The baseline RMT in the APB-triggered c-MNS was significantly higher than that in the BFT (*p* = 0.04).

Figure 2B shows the average MEP amplitude across all the participants. The two-way ANOVA showed a significant effect of time (F (3, 33) = 3.88, *p* = 0.018, η_p_^2^ = 0.261, 1 − β = 0.776), but no effect of intervention (F (2, 22) = 1.12, *p* = 0.35, η_p_^2^ = 0.092, 1 − β = 0.221) or interaction (F (6, 66) = 1.61, *p* = 0.159, η_p_^2^ = 0.128, 1 − β = 0.577). A post hoc analysis with Tukey’s test indicated that the MEP amplitude significantly increased at 0, 20, and 40 min post-intervention in the BFT (*p* < 0.05). In the APB-triggered i-TMS protocol, the MEP amplitude significantly increased at 0, 20, and 40 min post-intervention (*p* < 0.05). In the APB-triggered c-MNS protocol, the MEP amplitude significantly increased at 20 and 40 min post-intervention (*p* < 0.05) compared to that at baseline. The changes in the mean MEP amplitudes are shown in Figure 2B. For the MEP amplitude at baseline, a separate one-way ANOVA revealed no effect of intervention (F (2, 22) = 3.28, *p* = 0.06, η_p_^2^ = 0.23, 1 − β = 0.56).

### 3.2. SICI

The SICI group data are shown in Figure 3A. The two-way ANOVA yielded a significant effect of time (F (4, 44) = 3.84, *p* =0.009, η_p_^2^ = 0.259, 1 − β = 0.861), but no effect of intervention (F (2, 22) = 0.03, *p* = 0.98, η_p_^2^ = 0.002, 1 − β = 0.053) or interaction (F (8, 88) = 1.82, *p* = 0.08, η_p_^2^ = 0.142, 1 − β = 0.74). The results of the post hoc test showed that SICI significantly decreased at 0 min post-intervention in the BFT (*p* = 0.01). In the APB-triggered i-TMS protocol, SICI decreased significantly at 0 min post-intervention (*p* = 0.01) compared to that at baseline. There was no significant difference in SICI over time in the APB-triggered c-MNS protocol (*p* = 0.61).

### 3.3. ICF

The changes in ICF are shown in Figure 3B. The two-way ANOVA revealed a significant effect of time (F (4, 44) = 6.13, *p* = 0.01, η_p_^2^ = 0.358, 1 − β = 0.977), but no effect of intervention (F (2, 22) = 0.66, *p* = 0.57, η_p_^2^ = 0.057, 1 − β = 0.146) or interaction (F (8, 88) = 1.06, *p* = 0.396, η_p_^2^ = 0.088, 1 − β = 0.465). The post hoc test results showed that ICF significantly increased at 0 min post-intervention in the BFT (*p* = 0.02). In the APB-triggered c-MNS protocol, ICF increased significantly at 20 min post-intervention (*p* = 0.001). In the APB-triggered i-TMS protocol, there was no significant difference in ICF at different time periods compared to that at baseline (*p* = 0.09).

## 4. Discussion

In this study, we found that the MEP amplitude increased after all the tested bilateral movement protocols; BFT was considered the normal protocol, the APB-triggered c-MNS protocol used MNS, and the APB-triggered i-TMS protocol used TMS. Our results suggest that despite being voluntarily or artificially caused, repetitive bilateral movements cause long-lasting increases in motor cortical and corticospinal excitabilities.

### 4.1. Motor Cortical Excitability Changes Induced by Repetitive Apb-Triggered Stimulation

In the present study, the increase in the MEP amplitude lasted for up to 40 min in each intervention. The increase in the MEP amplitude may occur due to interactions in the motor cortex or subcortical structures [32]. The MEP may be increased because of primary mechanisms that increase the facilitatory circuits and/or decrease the inhibitory circuits in the M1. The RMT reflects the intensity of the stimulus that is needed to activate the most excitable corticospinal neurons and motoneurons and the RMT may fluctuate due to various variables. In our study, the RMT did not change throughout the experimental protocol. Hence, the influence of these elements is likely to be negligible. In our BFT protocol, ICF significantly increased, whereas SICI decreased. In a study by Waller et al., bilateral movement caused increased ICF and reduced SICI in both hemispheres [11]. These results are consistent with those of our study. Voluntary bilateral movement was considered to induce enhancement of the facilitation circuits and attenuation of the inhibitory circuits. In the APB-triggered c-MNS protocol, ICF significantly increased, but SICI did not change significantly. Conversely, the APB-triggered i-TMS protocol did not show a significant change in ICF but it showed a significant decrease in SICI. Based on this, it was considered that reducing the inhibitory circuit in APB-triggered i-TMS and enhancing the facilitatory circuits in the APB-triggered c-MNS protocol contributed to the increase in MEPs. There were no robust effects, such as those in the BFT protocol; however, changes in the cortical circuits might have occurred in both protocols.

Moreover, cortical excitability changes can occur with ES of the peripheral nerves and repetitive TMS (rTMS) alone. Previous studies have demonstrated that cortical plasticity can be induced by ES of the peripheral nerves. However, most previous studies on ES used high-frequency stimulation or train pulse stimulation. According to a review by Carson and Buick, the typical frequency of neuromuscular ES to activate sensory and motor axons should be 1–100 Hz [16]. Compared to those studies, the frequency that was used in our protocol was considered very small to have an impact on the brain.

As mentioned previously, rTMS causes brain plasticity. In particular, low-frequency rTMS (<1 Hz) causes depression in cortical activity [33]. The stimulation frequency that was used in our study was 0.2 Hz, and if the effect was due to rTMS, the MEP should have decreased. However, our results showed increased MEPs. Therefore, we speculated that a synergistic effect was observed when combining the input by interhemispheric communication from the contralateral motor cortex with the input via the sensory cortex by peripheral electrical stimulation or direct cortical stimulation by TMS.

### 4.2. Possible Neural Mechanisms

Short-term changes in the MEP have been induced by neuromodulation protocols such as rTMS, transcranial direct current stimulation (tDCS), and paired associative stimulation (PAS) [33,34,35,36]. tDCS is used to change the cortical excitability. Applying a weak electric current (1–2 mA) to the two electrodes that are placed on the skull causes depolarization during anodal stimulation and hyperpolarization during cathodal stimulation. In other words, anodal tDCS increases the excitability of the underlying cortex, whereas cathodal tDCS decreases it. The main factor of the long-term potentiation (LTP)-like outlasting neuroplastic mechanism of action of tDCS is thought to be the calcium-dependent synaptic plasticity of glutamatergic neurons [37]. PAS is a neuromodulation technique that uses two types of stimulation of the target cortical area. PAS is based on cellular-level experiments that reveal long-term depression (LTD) and LTP by spike timing-dependent plasticity, and the timing between the two stimulations should be more rigidly defined [36].

However, since our protocol used template-matching techniques for detecting muscle activity, we could not precisely control the interstimulus interval (the spike shapes that were detected in this technique did not always appear at the onset of muscle activity). Furthermore, in our study, spike timing was different for each protocol. The first stimulus was the signal when the motor command reached the contralateral motor cortex via the interhemispheric interaction, and this time was common in all protocols. In the APB-triggered i-TMS protocol, the second stimulation included the time from excitation of the motor cortex to the appearance of muscle activity, which approximated MEP latency (approximately 21 ms) and the mechanical delay of the spike detector [38]. In the APB-triggered c-MNS protocol, the total time included the time for peripheral nerve stimulation to reach the motor cortex via the sensory cortex (generally approximately 25 ms) in addition to the time that is mentioned above. Therefore, we were unable to discuss the time-locked effect on our results, as in the PAS protocol.

However, some studies have investigated non-time-dependent plasticity with multi-source inputs, including voluntary movement. Bisio et al. used paired stimulation protocols using voluntary finger movement and action that was observation with FES showed that the spontaneous movement tempo rate was significantly increased 30 min after the conditioning protocol [39]. A study by Bunday et al. that used paired corticospinal motoneuronal stimulation showed an increase in the MEP 30 min after intervention using TMS and peripheral nerve ES with voluntary movement [40]. These results suggest that inputs from multiple sources, including voluntary movements, may cause non-time-dependent changes.

### 4.3. Limitations and Future Directions

The present study has several limitations. The first limitation was that the statistical power did not reach a sufficient value (i.e., >0.8) in some statistical tests. This indicated the possibility that the statistical tests in the present study were not sensitive enough to detect a small difference (e.g., effect of intervention factor in ANOVA) due to the small sample size. A sufficient sample size might be needed to reduce type II errors in future studies.

The second limitation was the number of MEPs that were used to measure M1 excitability alterations. In the present study, the number of MEPs at each time point (i.e., 10 MEPs) was based on previous studies that suggested that 10 MEPs were required to obtain reproducible MEP amplitudes [28,29,30]. However, variability in the MEP amplitudes could be affected by various factors (such as target muscles, stimulus intensity, stimulus sites, inter-session intervals, and participant characteristics) [31,41]. Therefore, to measure a high number of MEPs to a realistic extent, it might be important to obtain reliable MEP amplitudes under various conditions.

The present artificial bilateral movement protocols and conventional BFT protocol showed a long-lasting increase in MEPs. These results provide evidence of increased M1 excitability in healthy subjects. Therefore, in the future, these protocols can be further investigated for designing new rehabilitation protocols for stroke survivors.

## 5. Conclusions

BMT is considered a useful method for increasing cortical excitability and preventing learning non-use of the paretic hand. However, it does not adapt to patients with severe paresis. Therefore, we examined the possibility of using BMT with artificial stimulation. Our results suggest that BMT with artificial stimulation increases cortical excitability, similar to that with conventional BMT. It may be possible to reduce the abnormal IHI by adding our technique when using it on the unaffected hand. However, it is difficult to explain the underlying mechanisms based on the results of our study.

There are some mechanisms, such as LTP-like long-lasting changes in cortical excitability, accumulation of frequent stimulation by rTMS and ES, time-dependent changes by PAS, and changes in the cell membrane by continuous energization by tDCS. However, our method is different from these protocols and it is presumed to be due to another physiological mechanism. Further research is needed to elucidate these mechanisms in the future.

## Figures and Tables

**Figure 1 brainsci-12-00061-f001:**
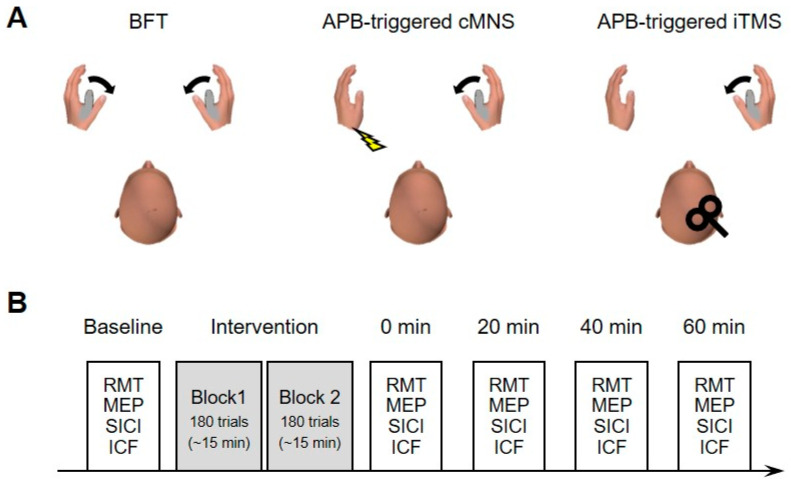
(**A**) Experimental protocol: Schematic illustration of the intervention protocols. (**B**) Time course of each experiment session.

**Figure 2 brainsci-12-00061-f002:**
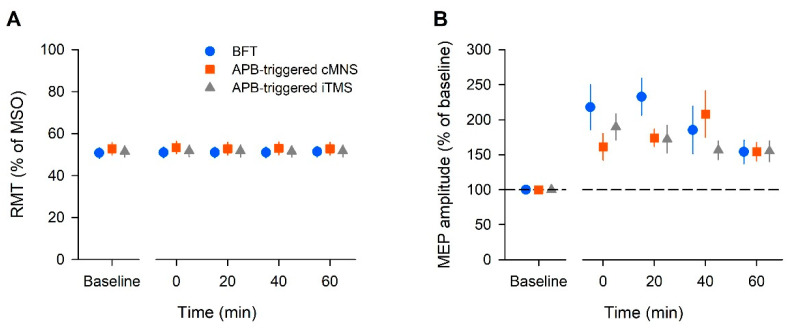
The averaged data of the resting motor threshold (RMT) (**A**) and amplitude of the motor-evoked potential (MEP) (**B**) in all the participants. The RMT is expressed as a percentage of the maximum stimulator output (MSO). The MEP amplitude is expressed as a percentage of the baseline value. Each plot and error bar represents the mean and standard error of the mean, respectively.

**Figure 3 brainsci-12-00061-f003:**
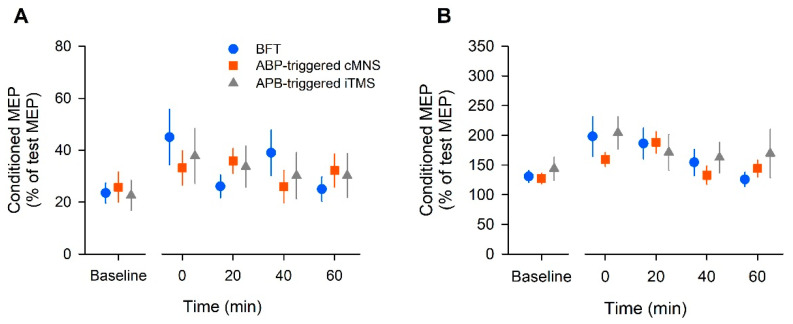
The population data of short-interval intracortical inhibition (SICI) (**A**) and intracortical facilitation (ICF) (**B**) in all participants. The conditioned MEP amplitude was expressed as a percentage of the unconditioned MEP amplitude. Each plot and error bar represents the mean and standard error of the mean, respectively.

## Data Availability

The data presented in this study are available on reasonable request from the corresponding author.

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
