# Peer review of "Motor Cortical Plasticity Induced by Volitional Muscle Activity-Triggered Transcranial Magnetic Stimulation and Median Nerve Stimulation"

_brainsci, 2021, doi:10.3390/brainsci12010061_

Round 1

Reviewer 1 Report

In the present study entitled ‘Motor cortical plasticity induced by volitional muscle activity-triggered transcranial magnetic stimulation and median nerve stimulation’, by Kariyawasam and colleagues, authors aimed to investigate the effects of artificial bilateral movement on corticospinal excitability through transcranial magnetic stimulation (TMS). For this purpose, motor-evoked potentials (MEPs) of primary motor cortex (M1) conventional bilateral motor training and artificial bilateral movements generated through electromyogram activity of the abductor pollicis brevis (APB) muscle triggered by peripheral nerve stimulation and TMS were recorded in 12 healthy participants. Three sessions with different interventions were conducted: bilateral finger training (BFT), right APB-triggered TMS of the ipsilateral M1 (i-TMS), and right APB-triggered contralateral median nerve stimulation (c-MNS). Results showed an increase in MEP amplitude up to 40 min post-intervention in all protocols compared to baseline and in all bilateral movement protocols.

In general, I think the idea of this article is really interesting and the authors’ fascinating observations on this timely topic may be of interest to the readers of Brain Sciences. However, some comments, as well as some crucial citations that should be included to support the authors’ argumentation, need to be addressed to improve the article, its adequacy, and its readability prior to the publication in the present form. My overall judgment is to publish this article after the authors have carefully considered my revisions below, in particular regarding the Introduction and Discussion section.

 Comments

  • Regarding the abstract: according to the Journal’s guidelines, authors should have provided an abstract of about 200 words maximum. Indeed, the current one includes 303 words. Please correct it.
  • In general, I recommend authors to use more references to back their claims, especially in the Introduction of the article, which I believe is lacking. Thus, I recommend the authors to attempt to deepen the subject of their manuscript, as the bibliography is too concise: nonetheless, in my opinion, less than 30 articles for a research paper are really insufficient. Currently, authors cite only 23 papers, and they are dramatically few, they should cite at least 40-50. Therefore, I suggest the authors to focus their efforts on researching relevant literature: I believe that adding more citations will help to provide better and more accurate background to this study. In this review, I will try to help the authors by suggesting relevant articles that suit their manuscript.
  • Page 1, Introduction: The introduction is overall well structured. However, in my opinion, a general overview of non-invasive brain stimulation (NIBS) commonly used to modulate brain activity and how they are applied to alter various cognitive domains (even clinical) is needed for non-expert readers. 
  • Page 1, Introduction: Authors stated that ‘Bilateral motor training (BMT) is a method used for improving post-stroke upper extremity function…that induces a short-term increase in corticospinal excitability after training. One of the hypothetical mechanisms of recovery with BMT is the normalization of abnormal interhemispheric inhibition’. In this respect, I believe that it may be useful to have more information from additional studies that explored the production and control of human movement and provided insights regarding neural and physiological processes that make motor control possible to help the rehabilitation of these skills, to help readers in gaining a deeper understanding of the topic: in a recent review, Borgomaneri and colleagues (2020, Cortex - https://doi.org/10.1016/j.cortex.2020.09.002) outlined how Non-invasive brain stimulation (NIBS) techniques can be used to alter the ability to manipulate prepotent ongoing motor actions in healthy individuals, defining the critical role of prefrontal areas, including the pre-supplementary motor area (pre-SMA) and the inferior frontal gyrus (IFG), in motor control. 
  • Page 2, Materials and Methods: I have a few concerns about the sample size, which is too small. This may reduce the power of the study, therefore I suggest reporting a power analysis that will determine the sample size that is most suitable to gain a level of significance. Also, subparagraphs indicating information about participants, study design, and Transcranial magnetic stimulation (TMS) should be indicated, so that the information can be properly understood by the readers and to allow the replicability of the study.
  • Page 7, lines 241-244: according to Authors, ‘The increase in MEP amplitude lasted for up to 40 min in each intervention. The increase in MEP amplitude may occur because of interactions in the motor cortex or subcortical structures. The MEP may be increased because of primary mechanisms that increase the facilitatory circuits and/or decrease the inhibitory circuits in M1’. In this regard, a study by Borgomaneri and colleagues (2021, Brain Sciences - https://doi.org/10.3390/brainsci11091203), might be of interest: here authors provided interesting insights on the use of single-pulse TMS to investigate the time course of the motor system readiness to relevant arousing stimuli (i.e., happy and fearful faces). Results showed an enhancement of corticospinal excitability (i.e., larger MEP) specifically in the early phase (i.e., 150 ms from picture presentation), addressing how early responses to emotional faces reflect enhanced motor readiness to arousing stimuli, hence, the preparation of adaptive motor responses required for the execution of appropriate behaviors.
  • Regarding the Conclusion: In my opinion, this section is too small and contains too many broad statements to adequately convey what the writers believe is the take-home message. To begin with, I would like to read here a brief summarization about studies that explored the application of bilateral motor training (BMT) and the use of electric stimulation in post-stroke treatment and how these techniques enhance corticospinal excitability, both voluntarily and artificially; therefore, I think that this section would benefit from more precise as well as in-depth considerations.
  • In according to the previous comment, I would ask the authors to include a ‘Limitations and future directions’ section before the end of the manuscript, in which authors can describe in detail and report all the technical issues brought to the surface.
  • Regarding the Figures and the Tables: please provide a short explanatory title for each figure and table in the main text. Also, I suggest modifying Figures 3 and 4 for clarity because, as it stands, the readers may have difficulty comprehending them. In my opinion, data settings are overcrowded and written with a very small font. I suggest to better organizing the graphs’ space in all the figures, to provide a better understanding and a direct interpretation of the
  • Page 8, Supplementary Materials: As a reviewer of this manuscript, I would have liked to check the effect sizes provided in Figure 3, but the dataset is not available. Please provide a working website.
  • The Reference list is incorrect: authors should check the Journal’s guidelines again and provide the abbreviated journal name in italics, the year of publication in bold, the volume number in italics.

Author Response

We would like to thank you for your thoughtful and insightful comments.
We modified manuscripts according to your comments and made a response letter to the reviewer. Please see the attachment.

Reviewer 2 Report

Introduction:

  1. In general, it is good to have one introductory para in which you explain what is happening after stroke and then go for some solution which might be e.g. BMT
  2. Also, it would be good if you first explain the interhemispheric inhibition mechanism and further explain how BMT results in normalization of abnormal interhemispheric inhibition.
  3. It might be useful to briefly introduce/explain the TMS and median nerve stimulation, mechanisms of action and their effective neural pass ways.
  4. It is also important briefly address what the SICI and ICF protocols are and what they measures.
  5. It should be clearer, why the author decided to measure the motor cortical excitability of contolateral hand!

Material and methods:

  1. There should be a rational for the number of participants, as, in the reviewer’s opinion, the study sample size is low. This should also be addressed at the limitation part of discussion.
  2. It is also important to explain in participants for the TMS and MNS safety related aspects? With relevant citation.
  3. How you identified ‘left APB’ hotspot. Please explain.
  4. What was the TMS coil angle
  5. In the reviewers’ opinion, 10 MEPs might not be sufficient for measuring motor-cortical excitability alteration. This is due to high variability of MEP amplitudes. Please justify, and also address this issue at the limitation part of the discussion section.
  6. For evaluating cortical excitability by TMS-MEP technique, the TMS coil was place over the left or right hemisphere?
  7. Did you include normalized baseline (1) into the ANOVA. If so, all ANOVAs should be repeated by excluding the baseline values and later you can compare after measures with baseline using student t-test. Also please include the level of each ANOVA factor e.g. intervention ‘3 levels’ and dependent, independent variables.
  8. Another ANOVA should be performed for baseline RMT measures, to see if there is a difference between measured RMTs across sessions.
  9. It might be useful to add jitter to the graph values (e.g. figure 3), as it is already really overlaid.

Discussion:

  1. Line 245: the argument that ‘RMT is one of the factors that change MEP’ is not correct.
  2. Line 250: ICI should be SICI. Please correct it.
  3. Line 260: rTMS is not defined before. Please correct it.
  4. It might be useful to investigate other modes of non-invasive brain stimulation, to manipulate cortical excitability e.g. tDCS.
  5. It might be important to include one section ‘limitation and future direction’. Please include the aforementioned limitations (sample size, low number of MEP measures…), as well as the transferability of your results to the clinical population. As one to one transferability is not granted.

Author Response

(The authors gave the same response as above.)

Round 2

Reviewer 2 Report

the authors have addressed all my concerns and I have no further comments.